# Bidirectional Underwater Drag Reduction on Bionic Flounder Two-Tier Structural Surfaces

**DOI:** 10.3390/biomimetics8010116

**Published:** 2023-03-11

**Authors:** Xixing He, Yihe Liu, Haiyang Zhan, Yahua Liu, Lei Zhao, Shile Feng

**Affiliations:** State Key Laboratory of High-Performance Precision Manufacturing, Dalian University of Technology, Dalian 116024, China

**Keywords:** bidirectional drag reduction, bionic flounder two-tier structures, numerical simulation

## Abstract

Engineering marvels found throughout the exclusive structural features of biological surfaces have given rise to the progressive development of skin friction drag reduction. However, despite many previous works reporting forward drag reduction where the bio-inspired surface features are aligned with the flow direction, it is still challenging to achieve bidirectional drag reduction for non-morphable surface structures. Inspired by the flounder ctenoid scales characterized by tilted, millimeter-sized oval fins embedded with sub-millimeter spikes, we fabricate a bionic *flounder* two-tier structural surface (BFTSS) that can remarkably reduce the forward skin friction drag by *η*_dr_ = 19%. Even in the backwards direction, where the flow is completely against the tilting direction of surface structures, BFTSS still exhibits a considerable drag reduction of *η*_dr_ = 4.2%. Experiments and numerical simulations reveal that this unique bidirectional drag reduction is attributed to synergistic effects of the two-tier structures of BFTSS. The array of oval fins can distort the boundary layer flow and mitigate the viscous shear, whilst the microscale spikes act to promote the flow separation to relieve the pressure gradient in the viscous sublayer. Notably, the pressure gradient relief effect of microscale spikes remains invariant to the flow direction and is responsible for the backward drag reduction as well. The bidirectional drag reduction of BFTSS can be extensively applied in minimizing the energy consumption of ships and underwater vessels, as well as in pipeline transport.

## 1. Introduction

Reducing skin friction drag is the crucial step for minimizing energy consumption and enhancing the working performance of ships and underwater vessels, as well as long-distance oil transport in pipelines [1,2]. The mainstream approaches in the quest for drag reduction include superhydrophobic surfaces [3,4,5], jetting microbubbles [6,7], drag-reducing polymer additives [8,9], and bionic surfaces [10,11,12]. Superhydrophobic surfaces can achieve excellent drag reduction by sustaining a plastron that gives rise to a large slip length at the interface, but the plastron tends to break away under high shear conditions, resulting in the failure or even the drag increase of superhydrophobic surfaces [13,14]. Jetting microbubbles is an active mechanism to reduce skin friction drag by injecting airflow into the surface to form a lubricating air layer, but it requires the mechanical installation of a bubble injection system and extra energy input [15]. In contrast, drag-reducing polymer additives focus on adding soluble long-chain polymers to the fluid flow to tune fluid properties and control the vortex structures [16,17]. However, it is usually restricted to flow in closed channels that does not entail the frequent addition needed to sustain a proper concentration of drag-reducing agents [18,19]. Using bionic surfaces is a promising passive approach in drag reduction and has received tremendous research attention, since it borrows the evolved surface structures and functions of living nature to alter the turbulent structure of the boundary layer flow [20,21,22].

During millions of years of evolution, aquatic creatures have leveraged finite resources to address the fierce survival challenges and have unlocked ingenious structures and functions for fast swimming with minimal drag [23,24]. Sharkskin covered with scales with microscale grooves can create low velocity vortices at the solid–liquid interface to reduce drag [25,26,27]. Pufferfish skin with adaptive stretching can reduce drag by slowing down the transition of the laminar flow state into turbulent flow state [28,29]. The ctenopharyngodon idellus can reduce surface friction drag by generating a fluid lubrication film which is attributed to the “water trapping effect” induced by the micro crescent unit [30]. 

Benefiting from the vast pool of aquatic creatures with exceptional underwater mobility, bionic surfaces have led to many design innovations in reducing skin friction drag [31,32,33]. Wang et al. [34] used laser etching to prepare a bionic tilted scale surface in one step and obtained a drag reduction of 4.8% by differential pressure measurement at Re = 2000. Zhang et al. [35] adopted a high-precision three-dimensional scanning method to reproduce shark skin and measured the highest drag reduction rate of 12.81% in the water tunnel test. Li et al. [36] prepared a flexible bionic shark skin using 3D printing technology and obtained a 6.6% increase in swimming speed and a 5.9% reduction in transportation cost in a robotic flapping experiment. Wu et al. [37] prepared a bionic surface with semi-circular grooves by milling, and measured an average drag reduction rate of 13% in the water tunnel experiment. Generally, the state-of-the-art design of bionic surfaces is characterized by tilted anisotropic surface protrusions. However, significant drag reduction can be achieved only if the flow is perfectly aligned with the tilting direction. Once the flow direction is reversed, the skin friction drag can even be increased. Therefore, it is still challenging to achieve bidirectional drag reduction for non-morphable surface designs [38]. 

Here, we report a two-tier structure of ctenoid scales of a flounder, i.e., tilted millimeter-sized oval fins embedded with sub-millimeter spikes at the outer edges. Inspired by this unique structure, we design and fabricate a bionic flounder two-tier structural surface (BFTSS) via 3D printing. BFTSS possesses bidirectional underwater drag reduction in turbulent conditions with water flow aligned with and against the tilting directions of surface structures, which exhibit the skin friction drag reduction rate *η*_dr_ measured by the pressure difference between the upstream and downstream of 19% and 4.2%, respectively. This unique bidirectional drag reduction is realized by the synergistic effects of the two-tier structures. For fluid flows over the designed surface, the first tier of oval fins can direct the fluid flow and reduce the energy loss associated with the viscous shear. The second tier of microscale spikes acts to facilitate the boundary layer flow separation and relieve the pressure gradient in the viscous sublayer. All these results are clearly demonstrated and visualized by numerical simulation. We envision that the unique bidirectional drag reduction can be applied to underwater navigation, anchors, and pipeline transportation. 

## 2. Bionic Sample Preparation

### 2.1. Bionic Prototype Analysis

We first anaesthetized a flounder with ether and cut a piece of flounder skin off from its back (5 cm × 5 cm), which was then immersed in 20 wt% sodium bicarbonate solution for 10 min to remove the mucus. After rinsing with distilled water, the skin was soaked in anhydrous ethanol for 5 min and sonicated (KQ-250DE, Kunshan Ultrasonic Instrument Co., Ltd., Kunshan, China) for 3 min to remove any impurities on the epidermis. All of the above methods complied with the Chinese Animal Protection Law and there was no cruel dissection or abuse.

The treated flounder skin and scales were observed under an ultra-deep-field microscope (VHX-900F, KEYENCE Co., Ltd., Osaka, Japan). As shown in Figure 1a, the scales are arranged in a staggered periodic pattern on the skin with a transverse distance *P* = 2.5 mm and a longitudinal distance *W =* 3 mm between adjacent scales. Each column of scales is shifted by *W*/2. As shown in Figure 1b, all flounder scales are tilted on the epidermis with a tilting angle *α* = 10°~20° and thickness *T* ≈ 90 μm. The ctenoid scale is characterized by a circular sector with a radius *R* ≈ 750 μm and a central angle *β* ≈ 110°. Each individual scale also has tiny spikes that are *L*_s_ ≈ 300 μm in length at the outer edges. The number of spikes *N* on each scale ranges from 1 to 9, but is mostly 5 or 6.

### 2.2. Sample Fabrication

We fabricated the bionic flounder two-tier structural surface (BFTSS) by using a 3D printing technique (Suzhou RayShape 3D Technologies Co., Ltd., Suzhou, China, Appendix A). It was printed by curing the resin layer-by-layer by exposure to 405 nm UV light, with the energy density, exposure time, exposure resolution, and layer thickness being 6 mW/cm^2^, 0.8 s, 25 μm, and 25 μm, respectively. The printing material was a flexible acrylic photosensitive resin “EL150”, also developed by Suzhou RayShape 3D Technologies Co., Ltd., Suzhou, China. The resin has the following parameters: 90 mPa∙s at 30 °C, a density (cured resin) of 1.09 g/cm^3^, a modulus of elasticity of 25 Mpa, an ultimate tensile strength of 6 Mpa, a tear strength of 15 N/mm, and a Shore A hardness of 80. As shown in Appendix A, the resin exhibits a hydrophobic property after curing and stabilization. The contact angle of 126° was measured by an angular contact meter (TBU 100, Dataphysics GmbH Co., Ltd., Stuttgart, Germany). 

The bionic prototype scale was simplified into a tilting and converging fin with a sector end with *N* (1 to 9) spikes. The detailed geometry is presented in Figure 1c,d. The fixed geometric parameters used in this study were the transverse distance *P* = 1.6 mm, the longitudinal distance *W* = 1.5 mm, the scale thickness *T* = 200 μm, the scale length *L* = 2 mm, the terminal radius *R* = 600 μm, and the central angle *β* = 120°. The central microscale spike has the largest length *L*_s_ = 300 μm, while the adjacent spikes are 30 μm shorter and vice versa. In this study, the number of spikes *N* and the tilting angle *α* (Appendix A) are varied to investigate the mechanism and performance of BFTSS in drag reduction. In order to distinguish the effect of microscale spikes, we also fabricated controlled samples without tiny spikes, i.e., cycloid scales for comparison. 

## 3. Methods and Results

### 3.1. Drag Reduction Measurements 

In this study, the drag reduction performance of samples was determined by measuring the differential pressure between the upstream and downstream flow past the sample, as shown in Figure 2a. A 180 mm × 20 mm sample was mounted onto a commensurate slot that was manufactured by a CNC mill on the bottom of a horizontal pipe, which had a 20 mm × 20 mm cross section. The two pressure sensors were located 300 mm far away from the front end and the back end of the sample, 160 mm away from the water inlet and outlet, respectively. Water (20 °C, *μ* = 1.0016 Pa∙s) was used as the working fluid and pumped out of a water tank by a water pump (Zhoushan Sensen Co., Ltd., Zhoushan, China). The water flow rate was controlled by a throttle valve and measured by a flow meter (Hangzhou Yikong Technologies Co., Ltd., Hangzhou, China) to derive the flow velocity. Two pressure transmitters (Shanghai Haofen Automation Technology Co., Ltd., Shanghai, China) were used to measure and record the differential pressure Δ*P* between the leading edge and the trailing edge of the sample after the water flow had been stabilized. The detailed experimental setup can be found in Appendix A. 

To mimic the hydrodynamic condition of flounders swimming at 3 km/h to 5 km/h undersea, the water flow velocity was fixed at 1 m/s. The Reynolds number was calculated to be Re = 19,920, rendering a turbulent duct flow. From the perspective of energy conservation, the skin friction drag manifests itself in the pipe flow experiment as a friction loss term. Based on Bernoulli’s equation for an incompressible flow in a horizontal pipe with a constant diameter, the drag reduction rate *η*_dr_ can be defined as Equation (1):(1)$$\eta_{\mathrm{dr}}=\frac{\Delta P_{\mathrm{smooth}}-\Delta P_{\mathrm{bionic}}}{\Delta P_{\mathrm{smooth}}}\times 100\%$$
where ∆*P*_smooth_ is the pressure difference over a smooth test surface and ∆*P*_bionic_ is the pressure difference over a bionic surface in the experiment. Note that a positive value of *η*_dr_ indicates a drag-reducing sample, while a negative value suggests that the sample leads to an increased drag otherwise.

### 3.2. Drag Reduction Performance of BFTSS

The drag reduction performance of BFTSS was investigated by varying the tilting angle *α* and number of spikes *N*. Figure 2b presents the comparison of the drag reduction performance of BFTSS and single-tier cycloid scales. Note that the BFTSS used in this series of experiments had a fixed number of spikes *N* = 5, corresponding to the most predominant flounder scales. At small tilting angles, both BFTSS and cycloid samples reduced the flow drag and their drag reduction performance was enhanced as *α* increased. This is because the tilting scales, with or without microscale spikes, are able to direct the chaotic turbulent boundary layer flow and tune the vortex structures to avoid extra eddy dissipations. The detailed mechanism will be discussed in the following section. As *α* approached 16°, BFTSS achieved an optimal drag reduction rate *η*_dr_ = 17.43%, a more than 200% increase compared to the maximum drag reduction rate *η*_dr_ = 5.54% at *α* = 14° for the single-tier cycloid scale sample. As the tilting angle continued to increase, *η*_dr_ decreased and eventually became negative. For BFTSS, *η*_dr_ became negative for *α* ≥ 24°, whereas the cycloid scales lost their drag-reducing function for *α* ≥ 18°. The above-mentioned improvements in drag reduction for BFTSS compared to the single-tier sample highlights the superiority of the tiny spikes of ctenoid scales in reducing drag for swimming flounders. The importance of incorporating two-tier surface features in designing drag-reducing surfaces is further demonstrated in Figure 2c. For different tilting angles, *η*_dr_ monotonically increased as the number of spikes increased from *N* = 0~9. With *N* = 9, the maximum drag reduction rate was further increased to 22.58% at *α* = 16°. In particular, for *α* = 18°, *η*_dr_ was initially negative for *N* = 0, corresponding to a single-tier cycloid scale sample, but a forward drag reduction (*η*_dr_ = 6%) was achieved once a single spike was added to each scale. Such an immediate transition in the drag reduction rate provides additional evidence for the dominant role of second-tier spikes in reducing the skin friction drag. 

### 3.3. Bidirectional Drag Reduction of BFTSS

The remarkably intriguing feature that we observed for BFTSS is its bidirectional drag reduction, in contrast to conventional structured surfaces that can only achieve drag reduction in the forward direction. Figure 2d shows the backward drag reduction performance of BFTSS at different *N* and *α*. For *N* = 0, i.e., the single-tier cycloid scales, it is found that the pressure loss always increased no matter how the titling direction *α* changed. For BFTSS with *N* < 7, we notice that although *η*_dr_ stayed negative, the pressure loss incurred by the sample actually decreased. A closer look at Figure 2d further reveals an astonishing transition from drag-increasing to drag-reducing that occurred for *N* ≥ 7 and *α* = 12°~14° in backward flow. For *N* = 9 and *α* = 14°, the drag reduction rate of BFTSS in backward flow is *η*_dr_ = 4.2%, comparable to the maximum drag reduction that can be achieved on single-tier cycloid scales. This phenomenon demonstrates that the effect of the second-tier spikes on BFTSS in reducing drag ought to be invariant to the flow direction and the unique bidirectional drag-reducing capabilities warrants that the application of BFTSS can be extended to more versatile settings. We also tested the effect of flow velocity on drag reduction by choosing BFTSS with *N* = 9 and *α* = 14°. As shown in Appendix A, drag reduction at low velocity is better than at high velocity. The maximum drag reduction rate *η*_dr_ = 27.74% was obtained at *u*_0_ = 0.839 m/s in forward flow. Additionally, the maximum drag reduction rate *η*_dr_ = 15.07% was obtained at *u*_0_ = 0.751 m/s in backward flow. Therefore, the effect of flow velocity on the drag reduction performance of BFTSS deserves more research attention in future.

## 4. Discussion

### 4.1. Numerical Simulations

In order to clarify the bidirectional drag reduction mechanism of BFTSS, we carried out a series of numerical simulations on Fluent to investigate the pressure and flow field development near BFTSS. In this study, a steady-state incompressible water flow (*ρ* = 998.2 kg/m^3^ and *μ* = 0.001 Pa∙s) was assumed and an RNG k-epsilon model was used to account for the turbulent duct flow. A three-dimensional computational domain simulating the experimental setup was modelled on SOLIDWORKS (Appendix A). It comprised a rectangular body with bionic fish-scale structures on the bottom surface and a smooth surface on the upper surface to facilitate a direct comparison of the results. We then meshed the computational domain on ICEM using a tetrahedral unstructured mesh capable of capturing the bionic structures. As shown in Appendix A, in order to obtain a more accurate simulation result, an inflation layer with an initial height of 3 μm, a growth rate of 1.2, and a layer number of 6 was added near the solid wall to capture the flow behaviors in the boundary layer. Furthermore, the mesh was refined near the bionic scales, generating 4.33 million cells in total. Water entered the computational domain with a uniform velocity *u*_0_ = 1 m/s through the inlet. The turbulence intensity was calculated by the following Equation (2):(2)I=0.16(Re)−1/8(Re=ρUDHμ,DH=4MX)
where *I* is the turbulence intensity, Re is the Reynolds number, *ρ* is the fluid density, *U* is the velocity of the fluid, *D*_H_ is the hydraulic diameter of the channel, *μ* is the dynamic viscosity of the fluid, *M* is the cross-sectional area of the rectangular channel and *X* is the perimeter of the cross-sectional area. The inlet was set to a velocity inlet with 5% turbulence intensity. A pressure outlet boundary condition was set for the outlet and no-slip boundary conditions were applied for the other walls. A second-order upwind scheme was used for discretizing the pressure and momentum and the solution was iterated in time using SIMPLEC algorithm and the convergence residuals were set to 1 × 10^−6^ for smaller tolerances. A new drag reduction rate *η*_s_ was defined by measuring the force change in simulations according to Equation (3):(3)$$\eta_{s}=\frac{F_{\mathrm{smooth}}-F_{\mathrm{bionic}}}{F_{\mathrm{smooth}}}\times 100\%$$
where *F*_smooth_ and *F*_bionic_ are the total forces acting on the smooth and bionic surfaces in the flow direction. The frictional force consists of a pressure contribution and a viscous contribution acting on the surfaces. Pressure force is calculated by integrating the pressure *p* over the sample surface and viscous force is the integral of the *x* component of wall shear stress *τ_x_* over the sample surface. The pressure force and viscous force can be calculated by Equations (4) and (5):(4)$$F_{\mathrm{pressure}}=\int_{A}^{}\tau_{x}\cdot n_{x}dA$$
(5)$$F_{\mathrm{viocous}}=-\int_{A}^{}p\cdot n_{x}dA$$
where *n_x_* is the *x* component of the unit outer normal vector of the sample surface *A*.

### 4.2. Drag Reduction Mechanisms of BFTSS 

Figure 3a shows the forward drag reduction rate *η*_s_ predicted by numerical simulations. Generally, the forward drag reduction performance of BFTSS was better than the single-tier cycloid scale surface. As *α* increased, *η*_s_ firstly increased and then dropped down below 0. The maximum drag reduction rates predicted by simulation *η*_s_ were 14.42% and 4.38%, respectively. In general, the simulation results are consistent with the experimental results. 

To further clarify the origin of drag reduction of BFTSS, we analyzed the viscous and pressure contributions of the flow drag. In this study, the samples were subject to a flow drag that can be further decomposed into the pressure force, resulting from the pressure difference between the front and back of tilted scales, and the viscous force, caused by shear stresses applied by stream. As the tilting angle *α* increased, the frontal area, i.e., the projection of scales perpendicular to the flow direction, increased, and consequently led to an increased pressure force. In addition, the flow stagnation caused by the protruded scales created local high-pressure points near the frontal surface of each scale and further amplified the pressure force. Therefore, a definitive increase in pressure force with respect to *α* was observed for both BFTSS and single-tier cycloid scales, shown in Figure 3b,c. The viscous force, however, actually decreased as *α* increased. This is because the existence of scales on the sample acts to increase the boundary layer thickness *δ* by distorting the boundary layer flow. As the velocity gradient can be scaled as $\frac{\partial u}{\partial y}\sim \frac{u_{0}}{\delta }$, the resultant shear stress $\tau =\mu \frac{\partial u}{\partial y}$ was also decreased. In Appendix A it is shown that when water encountered the scales, the velocity gradient in the near-wall region of the surface of either the BFTSS or the single-layer cycloidal scales was significantly smaller than that of the smooth surface. Moreover, compared to cycloid scales, BFTSS is better at influencing and reducing the velocity gradient of the back scales for less resistance. When *α* was very small, the pressure contribution to the drag was trivial. Therefore, the total drag decreased for increasing *α*. However, as *α* gradually increased, the pressure force dominated over the viscous force and the total drag increased with respect to *α*.

### 4.3. Effect of Second-Tier Spikes

A direct comparison between Figure 3b and c highlights the effect of the second-tier spikes in drag reduction. Apparently, the existence of second-tier spikes does not necessarily change the distribution of shear stresses in the boundary layer, as the viscous force for BFTSS was almost unchanged compared to that of single-tier cycloid scales. However, the growing rate of the pressure force for BFTSS was much slower. At *α* = 30°, the pressure force reached 11.2 × 10^−3^ N for cycloid scales whilst it was limited to only 8.8 × 10^−3^ N for BFTSS. The major difference in pressure force was attributed to the enhanced boundary layer flow separation caused by the spikes, which can be shown by the velocity field and vortex structure in Figure 3d,e. Compared to cycloid scales, the strong vortex in BFTSS proved a separated boundary layer flow, which prevented the locally low-velocity sub-layer from mixing with the high-velocity upper layers. As a result, a locally low-velocity region, indicated by the blue color, was present behind each column to counteract the flow stagnation on the frontal face of each scale and further relieve the pressure gradient along each scale. We notice that such a pressure gradient relief effect can be further enhanced by incorporating more spikes on each scale, but was eventually saturated for *N* ≥ 5, as shown in Figure 4a,b.

### 4.4. Origin of Bidirectional Drag Reduction

The numerical simulation results demonstrate that the unique bidirectional drag reduction of BFTSS indeed results from the capability of the secondary spikes in reducing the pressure gradient, and such a pressure gradient relief effect is invariant to the flow direction. Figure 4c presents the change in the pressure and viscous contributions to the total drag in backward flow. Apparently, the viscous force remains almost unchanged, which further confirms the negligible influence of the spikes on the viscous stress. However, the pressure force at *N* = 1 is increased from 5.4 × 10^−3^ N in forward flow to 7.2 × 10^−3^ N in backward flow. Such a significant increase in the pressure force can be explained by Figure 4d, where a strong vortex is produced near the frontal face of each scale. The large vortex further interacts with the inclined scale and forms a dead corner with high pressure underneath the scale, which eventually leads to the increased pressure force. Nevertheless, the pressure force can be decreased by integrating more spikes onto each scale. For *N* = 9, the pressure force is reduced to 4.75 × 10^−3^ N, a 40% decrease compared to that without spikes. Figure 4d also reveals that the low-pressure region behind the spikes significantly shrinks to reduce the overall pressure force acting on each scale. This phenomenon clearly demonstrates that the pressure gradient relief effect of the spikes stays valid even when the flow direction is completely reversed. 

## 5. Conclusions

In summary, we designed and fabricated a bionic flounder two-tier structural surface by borrowing inspirations from the sophisticated features of flounder ctenoid scales, which are characterized by tilted, millimeter-sized oval fins embedded with sub-millimeter spikes on the outer edge. In forward flow where the flow direction coincides with the titling direction of surface structures, the skin friction drag can be reduced by 19%, whereas the drag reduction rate of single-tier cycloid scale surfaces is limited to 4.38%. Most notably, the bidirectional drag reduction is achieved for BFTSS with a drag reduction rate of 4.2% in backward flow. In order to clarify the mechanism underlying the bidirectional drag reduction, we carried out numerical simulations to investigate the velocity and pressure field development. It is found that the drag reduction of BFTSS results from the synergistic effect of the first-tier oval fins that alters the boundary layer flow to reduce the shear stress, and the second-tier spikes that facilitate the flow separation to relieve the pressure gradient. In backward flow, the pressure gradient relief effect of the second-tier spikes remains unaffected and contributes to eventual drag reduction at high spike numbers. We envision that the exceptional drag reduction rate of BFTSS as well as its unique bidirectional drag reduction rate will be of great promise in the minimizing the energy consumption of ships, underwater vessels, and pipeline transport.

## Figures and Tables

**Figure 1 biomimetics-08-00116-f001:**
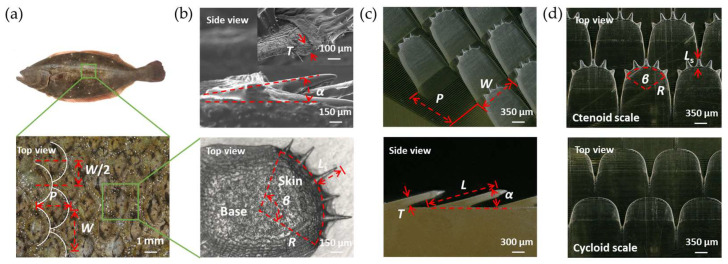
Flounder skin prototype morphology and bionic structure surface. (**a**) Flounder in full view and scales on the flounder skin. (**b**) Structural features of individual scales. (**c**) 3D view and side view of the bionic surface. (**d**) Structural features of ctenoid scale and cycloid scale.

**Figure 2 biomimetics-08-00116-f002:**
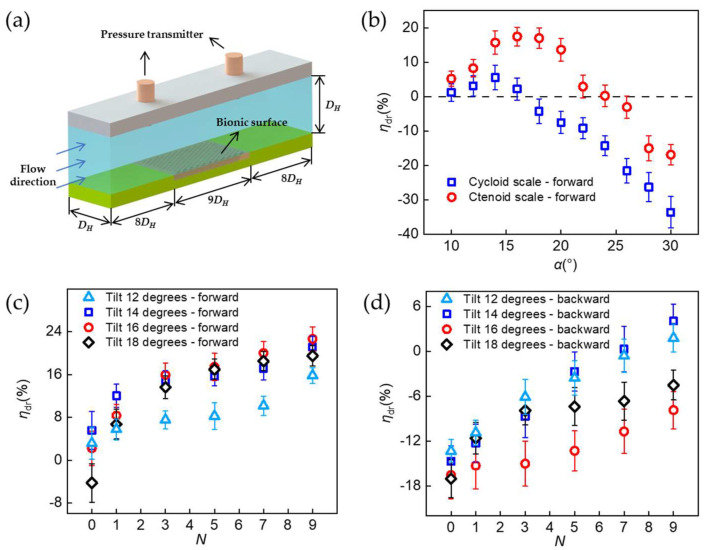
Drag reduction measurements. (**a**) Schematic of experimental set-up. (**b**) Forward drag reduction rates *η*_dr_ of ctenoid and cycloid scales at different tilting angle *α*. (**c**) Forward drag reduction rates *η*_dr_ at different number of spikes *N*. (**d**) Backward drag reduction rates *η*_dr_ at different number of spikes *N*.

**Figure 3 biomimetics-08-00116-f003:**
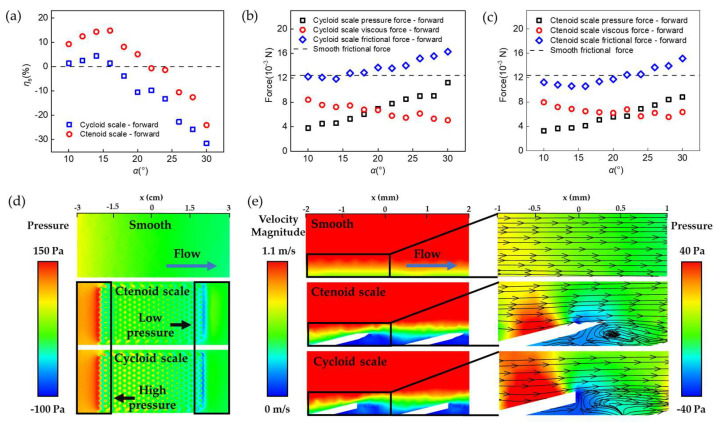
Numerical simulations of the drag reduction mechanism of BFTSS. (**a**) Simulated forward drag reduction rate for different surfaces with varying *α*. (**b**) Forces on single-tier cycloid scales with in forward flow. (**c**) Forces on BFTSS with in forward flow. (**d**) Pressure distribution on different surfaces in forward flow. (**e**) Velocity vector and streamline plots of different surfaces in forward flow.

**Figure 4 biomimetics-08-00116-f004:**
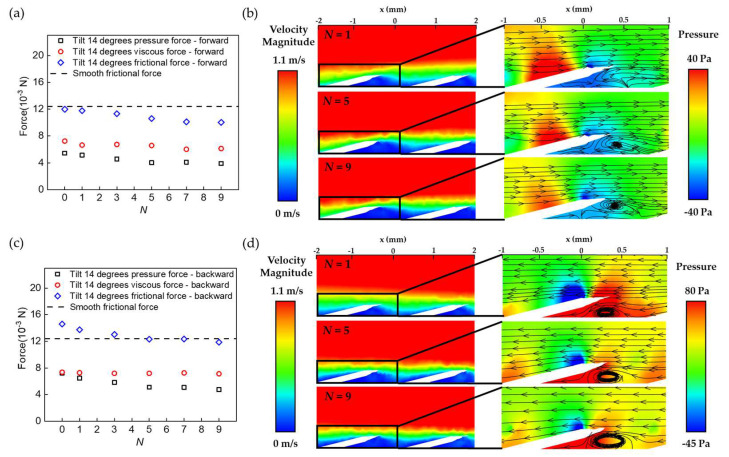
Mechanism of the bidirectional drag reduction. (**a**) The forces of BFTSS with different *N* in forward flow. (**b**) Velocity vector and streamline plots with different *N* in forward flow. (**c**) The forces of BFTSS with different *N* in backward flow. (**d**) Velocity vector and streamline plots with different *N* in backward flow.

## Data Availability

Not applicable.

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
