# Peer review of "Bidirectional Underwater Drag Reduction on Bionic Flounder Two-Tier Structural Surfaces"

_biomimetics, 2023, doi:10.3390/biomimetics8010116_

Round 1
Reviewer 1 Report
He et al. submitted a paper titled “Bidirectional Underwater Drag Reduction on Bionic Flounder two-tier structural Surfaces”. In this paper, inspired by the two-tier structural ctenoid scales of flounder, they manufactured a two-tier structural surface of tilted oval fins with spikes on outer margin through 3D printing to achieve the function of underwater bidirectional drag reduction. The key mechanisms for reducing the bidirectional resistance is that the array of oval fins can distort the boundary layer flow and mitigate the viscous shear, whilst the microscale spikes can promote the flow separation to relieve the pressure gradient in the viscous sublayer. This work is interesting and meaningful. Thus, I may recommend its publication as a full paper after addressing the following issues.
1. In the process of sample preparation, the printing parameters are not detailed.
2. Is the designed surface hydrophilic or hydrophobic? What is the contact Angle of the designed surface?
3. What is the effect of flow velocity on drag reduction?
4. How does the reverse drag reduction rate change with the tilt Angle?
5. The fish scale shown in Figure 1a has 6 spikes. Why did N=5 be selected in the experiment?
Author Response
We thank the referee for his/her appreciation of our work and we have significantly revised our manuscript as suggested. Please see the attachment aobut the Point-to-point response.
Reviewer 2 Report
Please see attached file.

Author Response
We thank the referee for his/her appreciation of our work and we have significantly revised our manuscript as suggested. Please see the attachment aobut the point-to-point response.
